# Effects on the Microstructure Evolution and Properties of Graphene/Copper Composite during Rolling Process

**DOI:** 10.3390/ma16165534

**Published:** 2023-08-09

**Authors:** Ziyue Yang, Fan Deng, Zhang Tao, Shuai Yan, Heng Ma, Miao Qian, Wei He, Zhifeng Zhang, Yanqiang Liu, Lidong Wang

**Affiliations:** 1Grinm Metal Composites Technology Co., Ltd., Beijing 101400, China; 2Zhejiang Huadian Equipment Testing Institute Co., Ltd., Hangzhou 310015, China; 3Wuhan NARI Limited Liability Company, State Grid Electric Power Research Institute, Wuhan 430074, China; 4School of Materials Science and Engineering, Harbin Institute of Technology, Harbin 150001, China

**Keywords:** graphene/Cu composite, cold rolling, interface, tensile property

## Abstract

Rolling treatments have been identified as a promising fabrication and deformation processing technique for graphene/metal composites with high performance. However, it is still a challenge to choose appropriate rolling parameters to achieve high strength, ductility and electrical conductivity of the composite simultaneously. In this study, graphene/Cu composites were prepared with an in situ growth method and rolling treatment. The effects of rolling deformation and temperature on the microstructural evolution of graphene and Cu grains, interface bonding between graphene and the matrix, mechanical and electrical properties were systemically investigated. The cold-rolled composite with 85% deformation displayed a maximum ultimate strength of 548 MPa, a high elongation of 8.8% and a good electrical conductivity of 86.2% IACS. This is attributed to oriented graphene arrangement and matrix grain refinement. Our research provides a comprehensive understanding for the rolling behavior of graphene/Cu composites, and can promote the development of graphene-based composites with high performance.

## 1. Introduction

Graphene, a two-dimensional nanomaterial consisting of single-layered carbon atoms, has aroused the interest of many researchers because of its unique properties, including its ultrahigh carrier mobility [1], superior thermal conductivity [2], distinct strength and Young’s modulus [3]. In recent years, graphene-reinforced metal matrix composites have gained increasing attention due to their remarkable structural and functional performance, and have shown an enormous potential for thermal management and electrical applications [4,5,6,7,8,9]. For instance, graphene/Cu composites-based heat sinks can be applied in smart electronic devices and high-performance computers [4], and tungsten–copper reinforced with graphene can be applied as electrical contacts in switching devices [5]. Moreover, high-quality graphene grown on Cu substrates or thin metal films via a chemical vapor deposition (CVD) method is also expected to be used in electronic devices [10,11].

The common problems that researchers face come into three aspects: (1) graphene easily aggregates in the metal matrix due to its huge surface energy and poor wettability to metals, which leads to overall performance degradation [12,13]; (2) the fabrication process may destroy the graphene microstructure, which directly causes a fall in the electrical or thermal properties of graphene or its composites [14,15,16]; (3) the adhesion between graphene and Cu is usually poor. Most researchers focus on solving the above difficulties, with the purpose of further enhancing the materials’ performances.

The rolling process is one of the most common plastic deformation methods used for metal matrix composites in the industry, and it has also been used recently in graphene/metal composites to improve their strength and toughness [17,18,19,20,21]. Mu et al. produced a graphene/Ti composite using spark plasma sintering and a subsequent hot rolling process, and the ultimate tensile strength of the composite with 0.1 wt.% graphene addition was found to be 54% higher than pure Ti [18]. Meng et al. constructed laminated graphene/Mg composite via spraying and a hot rolling technique. The ultimate tensile strength and elongation of the composite were increased by 32% and 12% with 0.75 vol.% graphene addition [22]. Wang et al. studied the hot rolling behavior of a 0.7 vol.% graphene/Cu composite, and speculated that the evolution of graphene during hot rolling underwent three processes: alignment along the rolling direction, fragmentation and exfoliation into thinner graphene nanosheets [23]. The composite with 80% rolling reduction rolled at 500 ℃ was shown to have a maximum ultimate strength of 441 MPa. Xiu et al. investigated the effects of rolling temperature and deformation on the graphene microstructure and composite performance in a 0.6 wt.% graphene/copper composite [24]. The results showed that high-temperature rolling contributes to avoiding damage to graphene during rolling and improving the properties of the composites. Besides graphene/metal composites, graphene also exhibits its own distinctive property in the case of deformation. Mescola et al. deposited graphene over different nanotextured silicon substrates and found that asymmetric straining of C-C bonds over the corrugated architecture resulted in distinct friction dissipation [25,26]. Thus, graphene deposited onto an anisotropic nanotextured system could acquire diverse nanomechanical properties, which have not been found in other 2D materials.

Although several successful examples using rolling deformation were found to improve composite properties, the fragmentation of graphene and the production of numerous residual stresses are still inevitable during the rolling process, which may lead to performance degradation. Therefore, a comprehensive investigation on the influence of rolling parameters on the microstructure evolution and properties of the composite is necessary. However, the previous research studies mainly focused on studying the effectiveness of hot rolling treatments on the enhancement of graphene/metal composite performance, the composite microstructure during cold rolling processes and the interface interactions between graphene and metal matrixes were not investigated deeply enough.

In this study, we use an ideal graphene/Cu composite fabricated with an in situ growth method from our previous study as a research model [27], in which graphene is uniformly distributed at the Cu grain boundary without introducing other influencing factors. We systematically investigated the microstructural evolution of the cold-rolled composites with different rolling deformations, which is lacking in previous research. The interface between graphene and the Cu matrix in the composites at different rolling temperatures was studied through X-ray photoelectron spectroscopy (XPS), and the mechanical and electrical properties of the composites were tested. This research provides a better understanding of the rolling behavior of graphene/Cu composites, and offers a novel strategy for the production of high-quality graphene and graphene-based composites.

## 2. Materials and Methods

### 2.1. Fabrication of the Graphene/Cu Composites

Cu powder was supplied by Ningbo Guangbo Nano Material Co., Ltd., with a purity of 99.9% and a particle size of about 300 nm. Graphene was synthesized using an in situ growth method from oleic acid (OA, over 99.7% purity, provided by Tianjin Guangfu chemical reagent factory (Tianjin, China)).

The preparation method of the graphene/Cu composite is described in ref. [27] and shown in Figure 1. Firstly, Cu powder was washed with 0.5 wt.% citric acid aqueous solution to remove the oxide on the Cu particle surface. Secondly, the etched Cu powder slurry was mixed with 0.5 wt.% benzotriazole (BTA) aqueous solution for 1 h, and then rinsed with ethanol 3 times to eliminate the excessive BTA. Thirdly, OA and Cu powder (mass ratio 1:100) were dispersed in the ethanol and were stirred for 1.5 h. The OA@Cu powder was obtained after drying in a vacuum at 80 °C. The OA coating Cu powder was consolidated with a spark plasma sintering method at a temperature of 600 °C and a pressure of 40 MPa for 5 min. The size of the as-sintered composite was 30 mm in diameter and 5 mm in thickness.

The rolling process was as follows: The as-sintered composite was annealed at a rolling temperature for 30 min. In order to compact the composite, the reduction of the first rolling pass was controlled to 40%. Since then, the rolling reduction was 5% during each rolling pass until reaching the final deformation. Then, the sample was annealed for 5 min at the rolling temperature before the next pass. The effects of the rolling temperature and reduction in the microstructure and properties of the composites were investigated. The rolling temperatures were set as 25 °C, 300 °C, 400 °C and 500 °C for comparison according to the recrystallization temperature of copper (269 °C). The rolling deformations of 30%, 55%, 70% and 85% were performed at 25 °C.

### 2.2. Characterizations

The morphologies of the composites were observed via scanning electron microscopy (SEM, Helios-Nanolab600i, Austin, TX, USA) and transmission electron microscopy (TEM, Talos-F200X, Troy, MI, USA). The mass fraction of graphene in the composite was 0.120 wt.%, as measured with a CS-901B carbon sulfur analyzer (supplied by Wanlianxinke Co., Ltd., Beijing, China). X-ray photoelectron spectra were obtained with a PHI 5700 ESCA System (supplied by Thermo Fisher Co., Ltd., Dreieich, Germany). A 5569R universal testing machine (supplied by Instron Co., Ltd., Norwood, MA, USA) was used to test the tensile properties of the graphene/copper composites. The tensile samples were cut along the rolling direction and had dog-bone shapes with a thickness of 1 mm and a width of 2 mm. In the tensile test, the displacement rate was 0.5 mm/min, and the gauge length was 15 mm. A 2512 current conductivity meter (supplied by Changsheng electronic technology Co., Ltd., Suzhou, China) was used for the electrical conductivity test.

## 3. Results and Discussion

### 3.1. The Effect of Rolling Deformation on the Cu Grain

Figure 2 shows that the representative EBSD images of as-sintered and cold-rolled composites after 30%, 55%, 70% and 85% reductions. It can be found that the as-sintered grains consist of isometric crystals. As the rolling reduction increases from 30% to 80% (Figure 2b–e), the grains are elongated gradually along the rolling direction and refined in the normal direction. The lengths of grains of different composites in the two directions (*d_RD_*) and (*d_ND_*) were measured. As shown in Figure 2f, the *d_RD_* and *d_ND_* are 0.68 mm and 0.63 mm, respectively. As the rolling reduction increases from 30% to 55%, the *d_RD_* increases from 0.96 mm to 1.47 mm, while *d_ND_* decreases from 0.50 mm to 0.28 mm, respectively. When the rolling reduction is more than 55%, both *d_RD_* and *d_ND_* undergo little change.

In order to achieve a better understanding of the role of graphene during the rolling process, TEM observations of the cold-rolled pure copper and composites with 85% rolling reduction were carried out (Figure 3). Figure 3a shows that the cold-rolled pure copper consists of 0.8–2 mm equiaxed grains. By contrast, the cold-rolled composite exhibits obviously anisotropic characteristics, and the grains have a length of 2–5 mm and a thickness of only 150–300 nm, which are stretched along the rolling direction, as shown in Figure 3b. A high-resolution TEM (HRTEM) image (Figure 3c) shows that the interface between the graphene nanosheets (GNS) and Cu matrix is smooth and clean, without any pores or cracks, and the graphene has a thickness of 8 nm. The morphological difference between the cold-rolled pure copper and graphene/Cu composites illustrates that the existence of graphene has a positive effect on the refinement of the matrix grains, which contributes to the improvement in the mechanical properties of the composites [28].

### 3.2. Effects of Rolling Temperature on Graphene Microstructure

In order to investigate the change rule of the microstructure and distribution of the graphene in the composite during the rolling process, XPS analysis and TEM observation were performed.

The XPS results of the composites rolled at different temperatures are shown in Figure 4 and Figure 5. The characteristic peaks of C_1s_, O_1s_, Cu_LMM_, Cu_2p3/2_ and Cu_2p1/2_ can be observed in the survey spectrum (Figure 4a). To further study the bonding configuration and defect state of graphene, the C 1s spectra of the composites are split into five peaks at 284.5 eV, 285 eV, 286.3 eV, 287.8 eV and 288.6 eV, which are assigned to the sp^2^ carbon, sp^3^ carbon, C-O, C=O and O-C=O bonds, respectively [29,30]. The proportions of the sp^2^ carbon, sp^3^ carbon and oxygen-containing groups in the composites were calculated by the ratio of their corresponding peak areas to the total C_1s_ peak area, as shown in Figure 4f. The content of sp^2^ carbon in the cold-rolled composite is 65%, which is lower than those in the hot-rolled composites. Accordingly, the contents of both the sp^3^ carbon and oxygen-containing groups decrease with increasing rolling temperature. It means that the hot-rolling process is beneficial to improve the quality of graphene in the composite. The reason may be that high rolling temperatures can accelerate copper surface catalysis and reduce graphene defects [31].

The O_1s_ peaks of the rolled composite were also fitted and split into three peaks including Cu-O-C (530.6 eV), C-O-C (531.6 eV) and C=O bonding (532.8 eV) [32], and their respective proportions are shown in Figure 5. It can be found that the content of Cu-O-C bonding (15%) in the low-temperature rolling composite is much lower than those in the high-temperature rolling composites (40–48%), while the content of C=O bonding shows an obviously opposite change rule. The percentage composition of Cu-O-C bonding slightly increases with elevated rolling temperature. Similar results were also reported previously through molecular dynamics (MD) simulations that showed a large number of covalent bonds in the graphene/Cu composite were generated during high-temperature hydrostatic compression due to the melting of copper nanoparticles [33].

According to the XPS results, the mechanism of the microstructural evolution of graphene with various rolling temperatures could be induced. The multi-pass rolling process makes the graphene fragment thin, and plenty of fresh graphene surfaces are generated. However, in situ grown graphene nanosheets have few oxygen-containing groups, and low-temperature rolling cannot provide a sufficient driving force to the formation of chemical bonding between graphene and Cu, but instead accumulate a lot of residual stress. By contrast, the hot-rolling process can not only prompt the conversion from C=O bonding to Cu-O-C bonding by thermal activation, but can also coordinate the deformation of graphene and relieve the fracture of Cu-O-C bonding by softening the matrix. It is reported that the Cu-O-C bonding contributes to improvement in the bonding strength between the graphene and copper matrix [34], which makes the load transfer from matrix to reinforcement effectively and results in increasing the strength of the composite. Therefore, increasing rolling temperature can improve the interface between graphene and the copper matrix.

The representative TEM images of the composites with different rolling temperatures are shown in Figure 6. It can be clearly observed that the grains in the cold-rolled composite are laminar, with a thickness of about 150 nm (Figure 6a). Moreover, the graphene nanosheets are homogeneously dispersed between the Cu matrix and along the rolling direction. By comparison, most grains are equiaxed in the hot-rolled composites, and several twin structures can be found in Figure 6b–d. As the rolling temperature rises from 300 °C to 500 °C, the average grain size increases from ~600 nm to ~1 μm. In addition, the graphene nanosheets move from the Cu grain boundary to the inner grain without changing the arrangement. The main reason for these phenomena is that the composites have poor graphene content (~0.49 vol.%), but go through severe deformation (~85%) along the rolling direction, which results in the graphene spreading out excessively along the RD direction; this cannot cover the Cu grains completely. Meanwhile, graphene nanosheets have poorer deformability compared with the copper matrix. The three-dimensional interpenetrating network structure of graphene formed during the SPS process is likely to break into plenty of fine and dispersed fragments due to significant deformation. That makes the Cu atoms on both sides of the graphene nanosheets diffuse. Obviously, a hot-rolling process can provide more driving force to promote the diffusion of Cu atoms than the cold-rolling process, which results in graphene sheets existing inside the Cu grain.

### 3.3. Mechanical and Electrical Properties of Composites under Different Rolling Parameters

Figure 7 shows the tensile properties of the as-sintered and cold-rolled composites. It can be seen that compared with the as-sintered composite, the ultimate strengths, yield strengths and elongations of the cold-rolled composites are all enhanced except the composite with 30% deformation, and even linearly increases with an increasing rolling ratio. In particular, the sample with the 85% rolling ratio has an ultimate tensile strength of 548 MPa and an elongation of 8.8%. The improvements in mechanical properties are mainly attributed to the refinement of the matrix grain, work hardening and the directional arrangement of the graphene nanosheets. Firstly, according to the above EBSD analysis as shown in Figure 1, a high rolling ratio contributes to the refinement of Cu grains, which enhance both the strength and plasticity of the composite. Secondly, the dislocation density of the copper matrix increases with increasing rolling deformation, and the dislocation tend to intercross and tangle each other, which further hinders the dislocation motion and increases the matrix strength. Moreover, the larger cold-rolling deformation is, the more well-aligned the graphene nanosheets at the Cu grain boundary are along the rolling direction, which makes the composite have a greater load capacity in this direction.

SEM observation was performed on the fracture surface of the cold-rolled graphene/Cu composites with different rolling ratios, as shown in Figure 8. There are obvious differences in the fracture mode between the small (30%, 55%) and large deformations (70%, 85%). When the rolling reduction is less than 55%, the morphological feature of fracture belongs to intergranular fracture (Figure 8a,b). Several particles (as indicated by the arrows) and microscopic cracks can be observed in the fracture surface. This indicates that the composites with low rolling deformation have poor interface bonding and are prone to interfacial debonding. When the rolling deformation exceeds more than 70%, the fracture surface presents typical ductile fracture features. Dimples and tearing ridges can be observed, and uncombined particles almost disappear. In addition, several pull-out graphene nanosheets can be found on the fracture surface, suggesting that the strong interfacial bonding between the graphene and copper matrix is beneficial to load transferring. The major reason is that some early micro-cracks are filled with matrix grains, and heal due to the rolling force. Previous molecular dynamics simulation results reported that the copper in the plastic flow state is constrained between graphene with a self-healing effect [35]. The self-healing effect of the copper matrix results in increasing density of the composite [36] and a good interface bonding between the graphene and copper matrix. Therefore, the strength and elongation of the composite with large rolling deformation were both improved, which is in accordance with the tensile test results.

The tensile properties of the as-rolled composites at different rolling temperatures are shown in Figure 9. It can be found that hot-rolled composites have higher elongations but lower strengths than the as-sintered composite. Both the yield and ultimate strengths of the composites decrease as the rolling temperature increases. Moreover, the elongation significantly rises by more than 8% compared with the as-sintered composite (~1%).

The impacts of the rolling temperature on the mechanical properties of the graphene/Cu composites can be attributed to grain refinement strengthening, grain boundary strengthening and work hardening. Firstly, the grain refinement strengthening mechanism can be expressed by the following:(1)$$\sigma =\sigma_{0}+k*d^{-1/2}$$
where σ is the yield strength of the composite, $\sigma_{0}$ is a frictional stress required to move dislocations, *k* is the Hall–Petch slope and *d* is the grain size. According to the above TEM results, the rolling temperature has a remarkable influence on the grain size of the Cu matrix, and obvious grain refinement can be achieved by the cold rolling treatment. Secondly, the rolling temperature also affects grain boundary strengthening in the composite. Graphene nanosheets are distributed at the grain boundary in the cold-rolled composite, but turn to become inside the grain in the hot-rolled composite. However, the image force is produced only when the graphene is distributed at the grain boundary, which can hinder the dislocation from slipping to the grain boundary and result in grain strengthening [27]. Furthermore, it has been reported that the segregation of the second phase at the grain boundary can reduce grain boundary energy and improve its stability [37]. The graphene in the cold-rolled composite, which is dispersed at the grain boundary, can stabilize the grain boundary and inhibit the dislocation motion. In addition, the work-hardening capacity of the matrix caused by different rolling temperatures is another factor of composite strength. The nominal rate of work hardening (Θ) can be expressed as follows:(2)$$\Theta =\frac{1}{\sigma }\frac{d\sigma }{d\varepsilon }$$
where σ is the true stress of the composite, and ε is the true strain of the composite. The calculated results of the composite at various rolling temperatures are shown in Figure 9b. It can be found that an increase in rolling temperature makes the nominal rate of work hardening reduce at the initial rolling stage, which leads to a decrease in the matrix strength. Previous MD simulation results also revealed that the graphene boundary could effectively enhance the strength of the strain-hardening rate of nanocrystalline metals [38]. Although hot rolling can effectively improve the quality of graphene and the interfacial bonding between graphene and the copper matrix compared with cold rolling on the basis of the XPS results, it still decreases the strength of the composite sharply. This suggests that refinement strengthening, grain-boundary strengthening and work hardening play important roles in the strengthening mechanism.

Figure 10 displays the fracture morphologies of the composites rolled at different rolling temperatures. Although all of the fractures of the as-rolled composites exhibit ductile fracture features, there are obvious differences between the cold-rolled and hot-rolled composites. The fracture surface of the cold-rolled composite exhibits the features of interlayer tearing and microscopic cracks. This suggests that interfacial debonding between graphene and copper matrix happens. When the rolling temperature rises to more than 300 °C, the fracture is characterized by well-developed dimples instead of interlayer tearing and cracks.

The electrical conductivities of the composite under different rolling parameters were tested, and the results are shown in Figure 11. The test direction is along the rolling direction. It can be seen that the composites with small rolling ratios (30% and 55%) have lower electrical conductivities than the as-sintered composite. When the rolling deformation is more than 70%, the electrical conductivity of the composite achieves a slight improvement. The conductivity of the rolled composite with 85% deformation is as high as 82.2% IACS. The main reasons for this are related to the initiation and healing of cracks, and the alignment of graphene nanosheets during the rolling process. Several microscopic cracks are generated at the interface between the graphene and the matrix in the initial stage of cold rolling. These cracks cause the scattering of free electrons, which results in a decrease in the conductivity. When the rolling ratio further increases, matrix particles fill in and may heal the crack [39]. Meanwhile, the graphene nanosheets in the composite tend to be arranged along the rolling direction. The two above factors both contribute to improvements in conductivity.

In addition, the conductivity of the composite slightly improved from 82.2% IACS to 86.2% IACS when the rolling temperature was increased from 25 °C to 500 °C. On the one hand, the original fine grain microstructure became coarser with increasing rolling temperature according previous TEM observations, so the number of grain boundaries decreases, which reduces the scattering of free electrons. On the other hand, based on the XPS results, graphene as a conductive functional medium in the composite has fewer defects at higher rolling temperatures, which is also an important positive factor on the improvements in conductivity.

It is worth mentioning that the cold-rolling treatment in this research with large deformations exceeding 80% surprisingly contributes to improve the mechanical and electrical properties of graphene/copper composites. However, significant performance degradation occurs in the cold-rolled composite fabricated with ball milling, with similar deformation [24]; the hardness of the cold-rolled composite with 80% deformation is only 105 HV, which is much lower than that in this study (162 HV). We can understand the phenomenon according to the differences in the interactions between graphene and the Cu matrix. In our study, when the rolling reduction was more than 80%, deformation of the Cu grains did not easily occur due to high density dislocation pile-up. This results in the graphene/Cu interface becoming the weakest part in the composite. Thanks to the in situ growth method, strong chemical bonds (Cu-O-C bonding) still exist between the graphene and copper grains, although the composite suffers from severe deformation, and the graphene can coordinate with the deformation of Cu grains during cold rolling. By contrast, it is difficult for simple mechanical mixing techniques to form good interface bonds between graphene and copper. Thus, cracks and holes easily result when the rolling deformation is too large, resulting in a significant decrease in the properties of the composite.

## 4. Conclusions

In this study, the evolution of the microstructure and properties of graphene/Cu composites under different rolling temperatures and rolling deformation were systematically investigated. A hot rolling treatment can reduce the defects of graphene and improve the interface bonding between graphene and a copper matrix, but deteriorate the strength of the composites due to the fragmentation of graphene that is swallowed by copper grains. For cold rolling, the mechanical and electrical properties of the composites both increase continuously with increasing rolling deformation. The cold-rolled composite with 85% deformation achieved a maximum ultimate strength of 548 MPa, a high elongation of 8.8% and a good electrical conductivity of 86.2% IACS. Our research provides a comprehensive understanding for the rolling behavior of graphene/Cu composites, and can promote the development of graphene-based composites with high performance.

## Figures and Tables

**Figure 1 materials-16-05534-f001:**
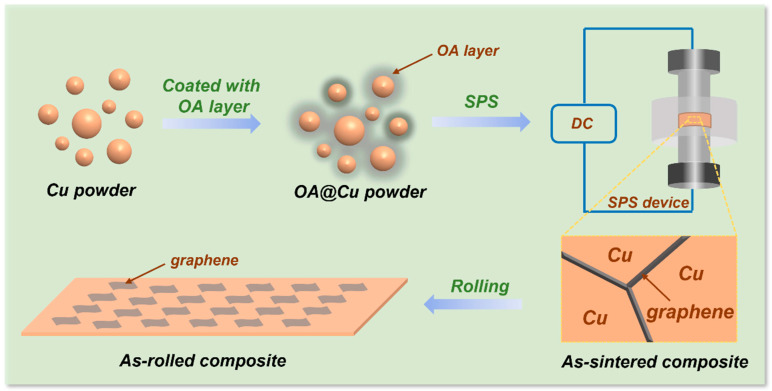
Schematic diagram of the preparation of graphene/Cu composite.

**Figure 2 materials-16-05534-f002:**
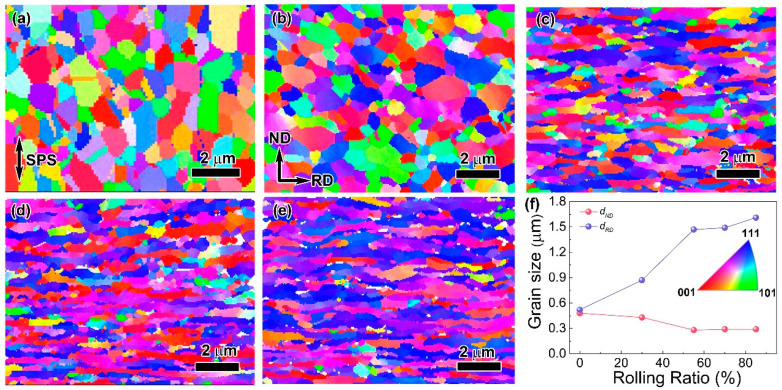
EBSD images of as-sintered and cold-rolled graphene/Cu composites with different rolling ratios: (**a**) as-sintered, (**b**) 30%, (**c**) 55%, (**d**) 70%, (**e**) 85% and (**f**) grain size distribution.

**Figure 3 materials-16-05534-f003:**
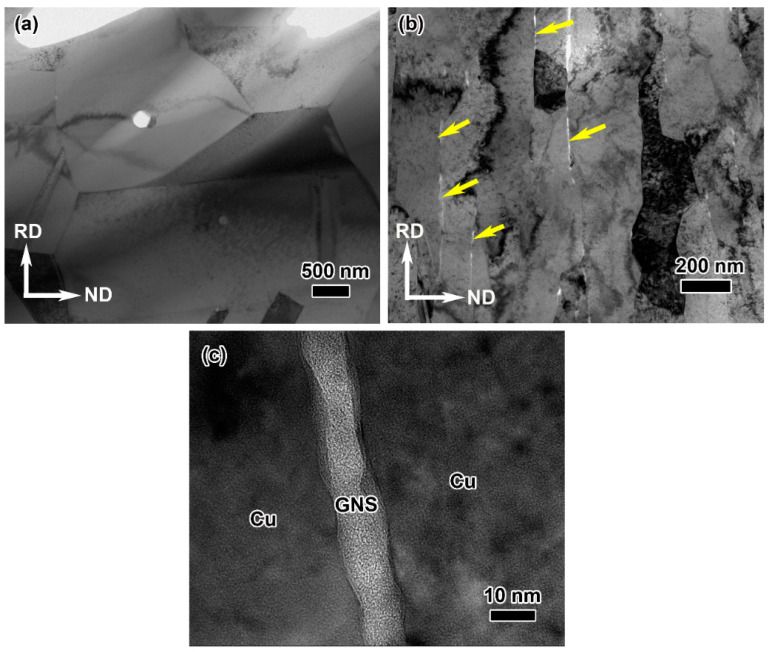
Microscopic morphology of cold-rolled pure copper and graphene/Cu composite. (**a**) TEM image of the cold-rolled pure Cu sample, (**b**) TEM and (**c**) HRTEM images of the cold-rolled graphene/Cu composite. The locations of the yellow arrows in (**b**) show the graphene in the composite.

**Figure 4 materials-16-05534-f004:**
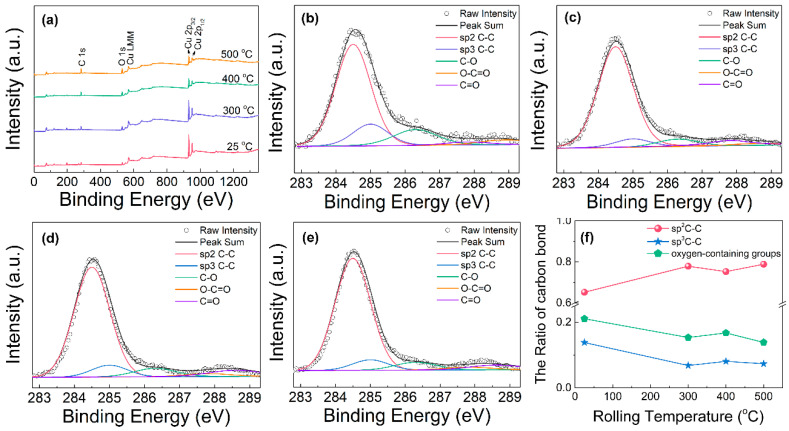
XPS analysis of the graphene/Cu composites rolled at different temperatures: (**a**) survey spectrum, C_1s_ spectrum at (**b**) 25 °C, (**c**) 300 °C, (**d**) 400 °C, (**e**) 500 °C and (**f**) the ratio of carbon bond.

**Figure 5 materials-16-05534-f005:**
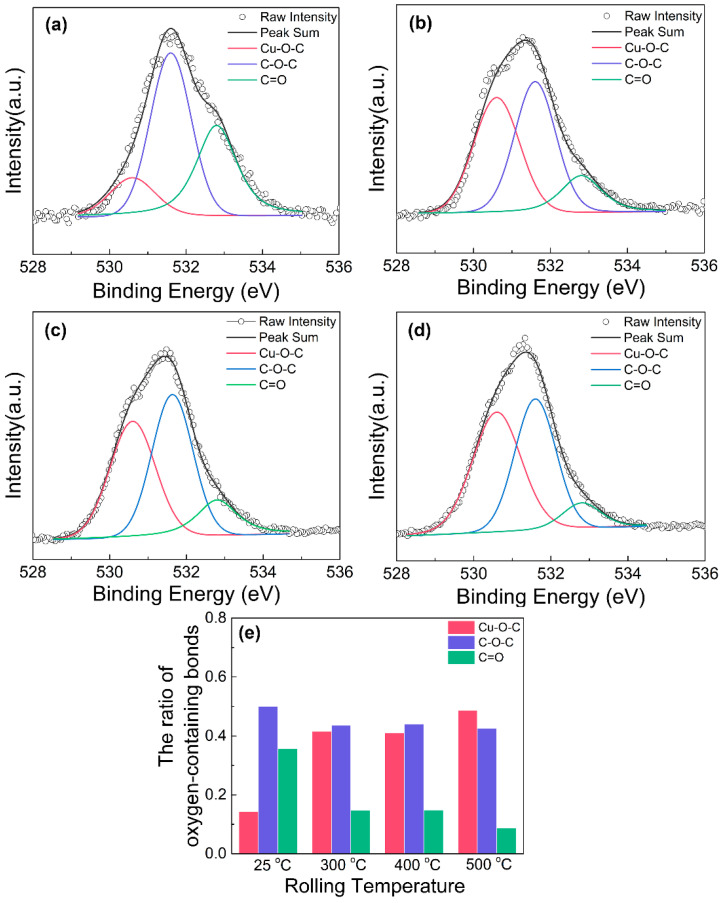
Curve fit of O_1s_ spectra of the graphene/Cu composites rolled at various temperatures: (**a**) 25 °C, (**b**) 300 °C, (**c**) 400 °C, (**d**) 500 °C and (**e**) the ratio of oxygen-containing bonds.

**Figure 6 materials-16-05534-f006:**
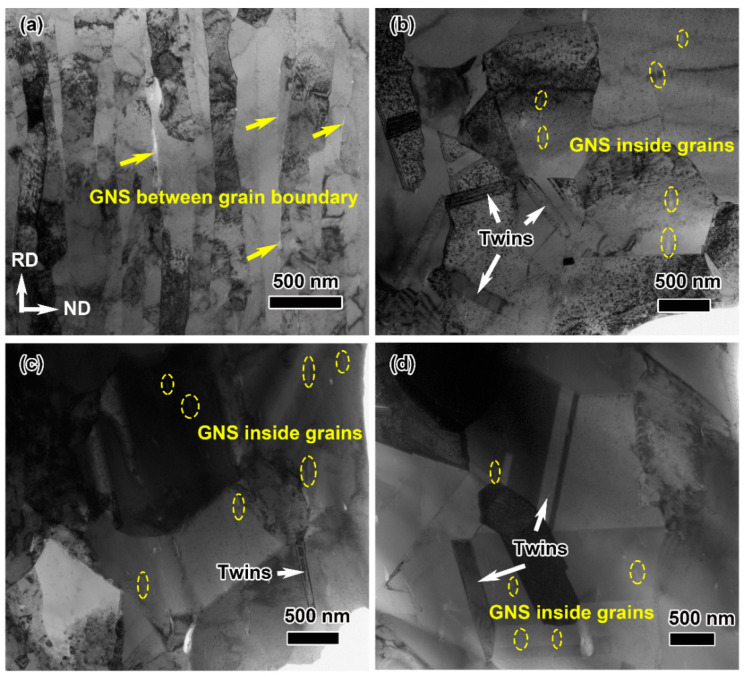
TEM images of graphene/Cu composites rolled at different rolling temperatures: (**a**) 25 °C, (**b**) 300 °C, (**c**) 400 °C and (**d**) 500 °C. The locations of the yellow arrows in (**a**) and yellow dashed circles in (**b**–**d**) show the graphene, and the white arrows mark Cu twins in the composite.

**Figure 7 materials-16-05534-f007:**
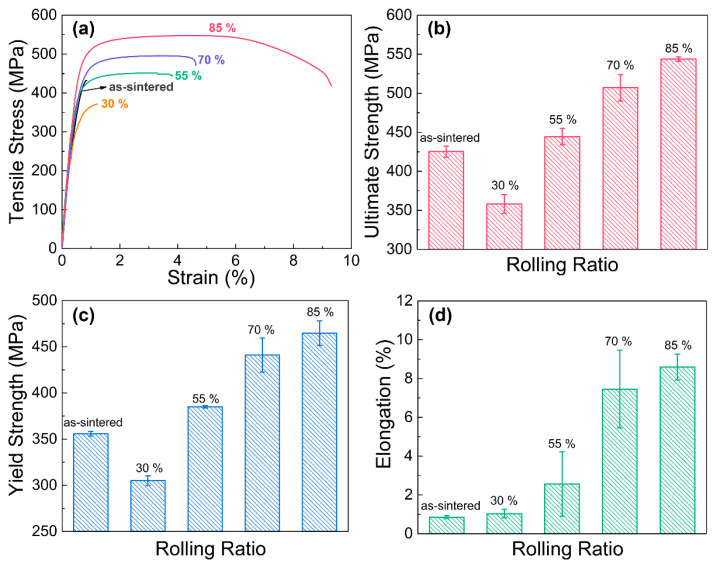
Tensile properties of the cold-rolled graphene/Cu composites with different rolling ratios: (**a**) typical tensile curves, (**b**) ultimate tensile strength, (**c**) tensile yield strength and (**d**) elongation.

**Figure 8 materials-16-05534-f008:**
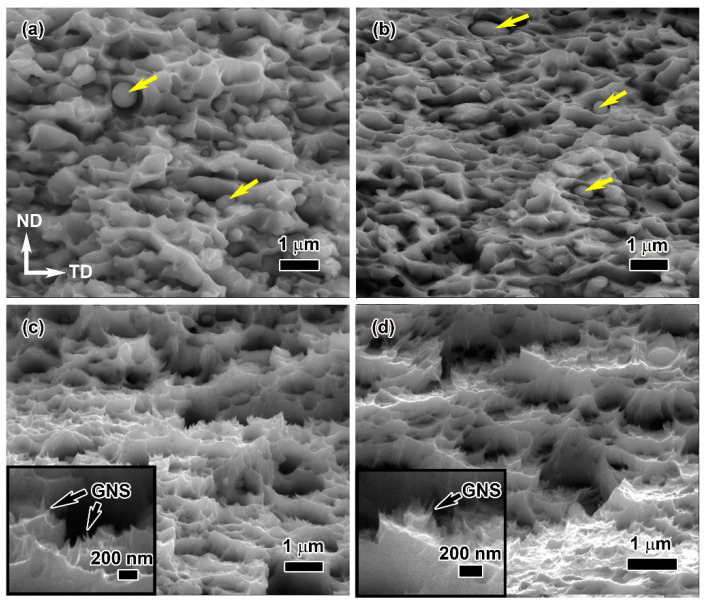
SEM images of fracture morphologies of the cold-rolled graphene/Cu composites with different rolling ratios: (**a**) 30%, (**b**) 55%, (**c**) 70% and (**d**) 85%. The locations of yellow arrows in (**a**) mark uncombined particles.

**Figure 9 materials-16-05534-f009:**
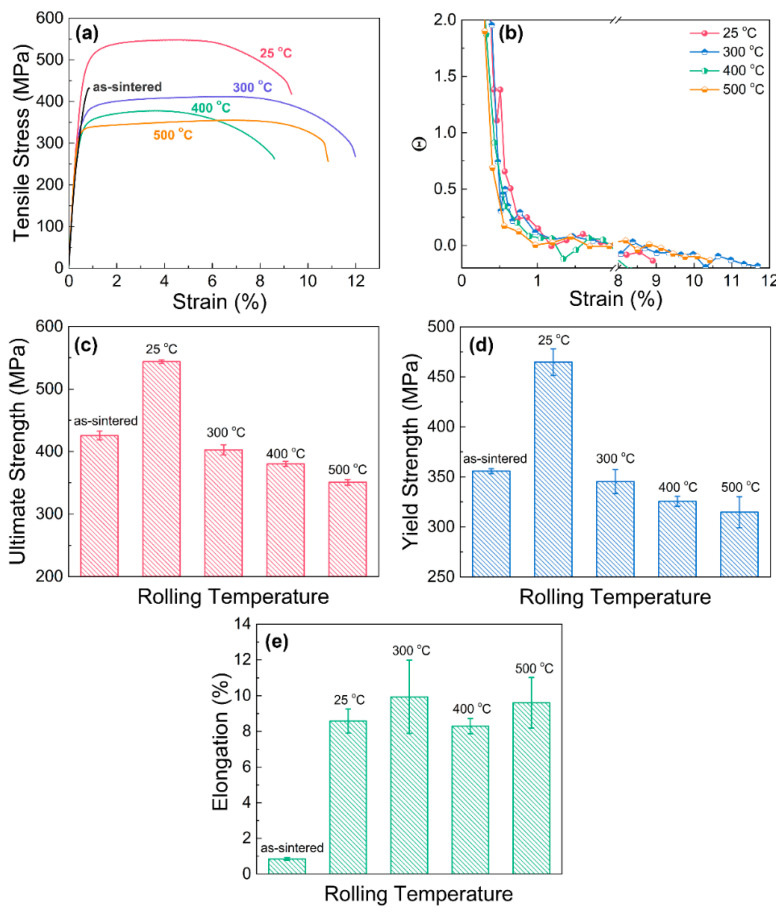
Tensile properties of the rolled graphene/Cu composites at various temperatures: (**a**) tensile curves, (**b**) normalized strain hardening rate (Θ) against true strain, (**c**) ultimate tensile strength, (**d**) tensile yield strength and (**e**) elongation.

**Figure 10 materials-16-05534-f010:**
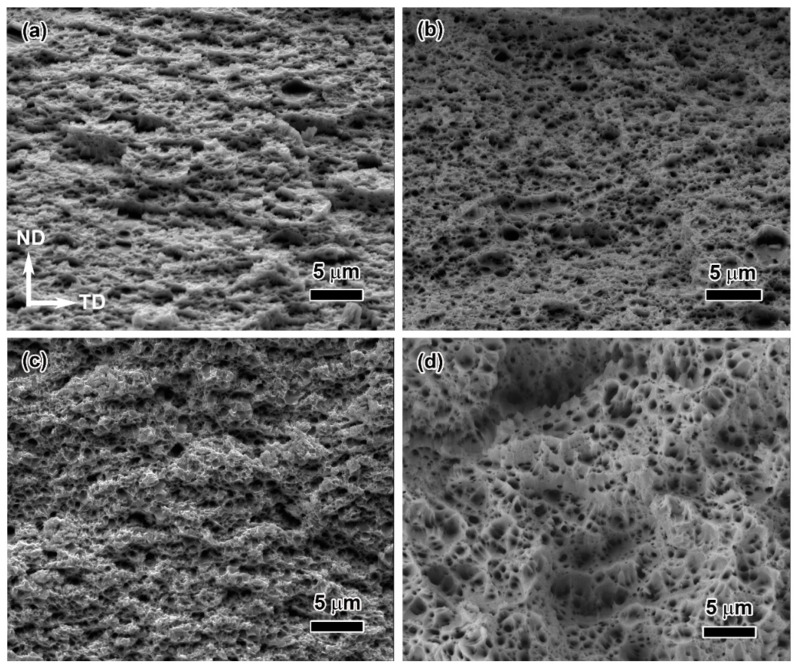
Fracture morphologies of the graphene/Cu composites rolled at various temperatures: (**a**) 25 °C, (**b**) 300 °C, (**c**) 400 °C and (**d**) 500 °C.

**Figure 11 materials-16-05534-f011:**
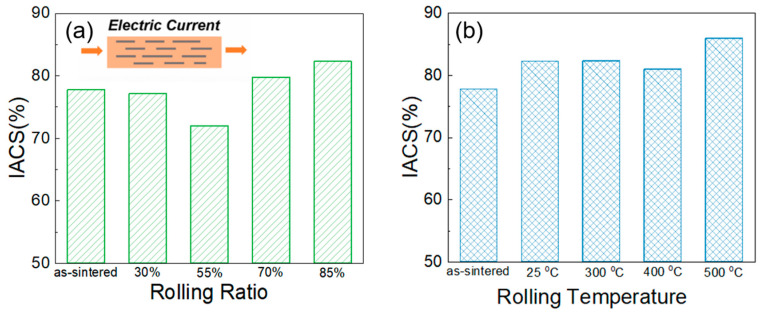
Electrical conductivities of the graphene/Cu (**a**) with different rolling ratios at room temperature and (**b**) under different rolling temperatures.

## Data Availability

The data presented in this study are available on request from the corresponding author.

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
