# Peer review of "Effects on the Microstructure Evolution and Properties of Graphene/Copper Composite during Rolling Process"

_materials, 2023, doi:10.3390/ma16165534_

Round 1

Reviewer 1 Report

Review Report

The authors investigated the effects of deformation and temperature on the microstructure evolution of graphene and Cu grains, the interfacial bond between graphene and matrix, and mechanical and electrical properties. This article is covered by the "Materials" journal. However, the article will be ready for publication after a major revision. Comments are listed below.

·         The introductory part is insufficient. It should be expanded by giving current references from the literature.

·         Multiple references given in the introduction are not recommended (....cable applications 30 [1-6], .....and toughness [12-16]).

·         The disadvantages of the Rolling process should be mentioned in the introduction.

·         More information about graphene should be given in the introduction.

·         What is the novelty of this article? The difference from similar studies in the literature should be clearly demonstrated.

·         According to which standards were the selected rolling temperatures (25 °C, 300 °C, 400 °C and 500 °C) determined?

·         Properties such as the dimensions, purity, and weight ratios of copper and graphene used in the material and method section should be given.

·         Giving an experiment schema in the Experimental section is recommended.

·         The parameters used in the tensile test should be given in the experimental section.

·         The resolution of Figure 1 should be increased.

·         The Results section should be discussed further. It should be discussed by comparing it with similar studies in the literature.

·         The article contains numerous typographic and language errors. It should be corrected.

·         The article should be rearranged by taking into account the journal writing rules and citation rules.

Author Response

Dear editors and reviewers: 

    Thank you for your letter and the comments concerning our manuscript entitled “Effects on the microstructure evolution and properties of graphene/copper composite during rolling process” (ID: materials-2525325). Those comments are all valuable and very helpful for revising and improving our paper, as well as the important guiding significance to our researches. We have studied the comments carefully and have made corrections which we hope meet with approval.

    Revised portions are marked in red in the revised manuscript. The main corrections in the revised manuscript and the responds to the reviewer’s comments are as follows:

Comment 1: The introductory part is insufficient. It should be expanded by giving current references from the literature.

Reply 1: Thanks for the reviewer’s suggestion.

    According to the reviewer’s comment, we expanded the introductory part by citing several current references.

    In the first paragraph, we cited two references to illustrate the application of high-quality graphene. Specific modifications are as follows:Moreover, high-quality graphene grown on Cu substrate or thin-metal films by chemical vapor deposition (CVD) method is also expected to be used in electronic devices [10,11].

   In the third paragraph, we cited two references [25,26] to explain the benefit of using graphene for the composite with respect with other 2D materials. Specific modifications are as follows:Besides graphene/metal composites, graphene also exhibits its own distinctive property in the case of deformation. Mescola et al deposited graphene over different nanotextured silicon substrates and found that asymmetric straining of C-C bonds over the corrugated architecture resulting in distinct friction dissipation [25,26]. Thus, graphene deposited onto an anisotropic nanotextured system could acquire diverse nanomechanical properties, which haven’t been found in other 2D materials.

The above discussion is added on Page 3 and 4 in the revised manuscript. Some references for above discussion are listed as follows:

[10] Frank, O.; Vejpravova, J.;Holy, V.; Kavan, L.; Kalbac, M., Interaction between graphene and copper substrate: The role of lattice orientation. Carbon 2014, 68, 440-451.

[11] Xu, S.; Zhang, L.; Wang, B.; Ruoff, R. S., Chemical vapor deposition of graphene on thin-metal films. Cell Reports Physical Science 2021, 2, (3).]

[25] Mescola, A.; Paolicelli, G.; Ogilvie, S. P.; Guarino, R.; McHugh, J. G.; Rota, A.; Iacob, E.; Gnecco, E.; Valeri, S.; Pugno, N. M.; Gadhamshetty, V.; Rahman, M. M.; Ajayan, P.; Dalton, A. B.; Tripathi, M., Graphene Confers Ultralow Friction on Nanogear Cogs. Small 2021, 17, (47), e2104487.

[26] Mescola, A.; Silva, A.; Khosravi, A.; Vanossi, A.; Tosatti, E.; Valeri, S.; Paolicelli, G., Anisotropic rheology and friction of suspended graphene. Physical Review Materials 2023, 7, (5).

Comment 2: Multiple references given in the introduction are not recommended (....cable applications 30 [1-6], .....and toughness [12-16]).

Reply 2: Thanks for the reviewer’s suggestion.

According the reviewer’s comment, we added the recommendation for some references in the introduction. Specific modifications are as follows:

…electrical applications [4-9]. For instance, graphene/Cu composite based heat sinks can be applied in smart electronic devices and high-performance computer [4] and tungsten-copper reinforced with graphene can be used as electrical contacts applied in switching device [5].

… to improve their strength and toughness [17-21]. Mu et al produced graphene/Ti composite by spark plasma sintering and subsequent hot-rolling process and the ultimate tensile strength of the composite with 0.1 wt.% graphene addition was 54 % higher than pure Ti [18].

The above discussion is added on Page 3 and 4 in the revised manuscript.

Comment 3:   The disadvantages of the Rolling process should be mentioned in the introduction.

Reply 3: Thanks for the reviewer’s suggestion.

According the reviewer’s comment, we added the disadvantages of the Rolling process on Page 5 in the revised manuscript. Specific modifications are as follows: Although several successful examples using rolling deformation achieves the improvement of composite properties, the fragmentation of graphene and the production of numerous residual stresses are still inevitable during the rolling process, which may lead to performance degradation.

Comment 4:   More information about graphene should be given in the introduction.

Reply 4: Thanks for the reviewer’s suggestion.

According to the reviewer’s comment, we added more information about graphene in the introduction.

Graphene, a two-dimensional nanomaterial consisting of single layered carbon atoms, have aroused many researchers’ great interest because of its unique properties including ultrahigh carrier mobility [1], superior thermal conductivity [2], instinct strength and Young’s modulus [3].

The above information about graphene is added on Page 3 in the revised manuscript.

Some references for above discussion are listed as follows:

  1. Zhang, Y. B.; Tan, Y. W.; Stormer, H. L.; Kim, P., Experimental observation of the quantum Hall effect and Berry's phase in graphene. Nature 2005, 438, (7065), 201-204.
  2. Balandin, A. A.; Ghosh, S.; Bao, W. Z.; Calizo, I.; Teweldebrhan, D.; Miao, F.; Lau, C. N., Superior thermal conductivity of single-layer graphene. Nano Letters 2008, 8, (3), 902-907.
  3. Lee, C.; Wei, X. D.; Kysar, J. W.; Hone, J., Measurement of the elastic properties and intrinsic strength of monolayer graphene. Science 2008, 321, (5887), 385-388.

Comment 5:   What is the novelty of this article? The difference from similar studies in the literature should be clearly demonstrated.

Reply 5: Thanks for the reviewer’s suggestion. According to the reviewer’s comment, we rephrased the novelty of our article as follows.

We systematically investigated on microstructure evolution of the cold-rolled composites with different rolling deformations which is lack of research before. The interface between graphene and Cu matrix in the composites at different rolling temperatures was studied through X-ray photoelectron spectroscopy (XPS) and the mechanical and electrical properties of the composites were tested. This work would provide a better understanding of the rolling behavior of graphene/Cu composite and offer a novel strategy for the production of high-quality graphene and graphene-based composite.

The above discussion is added on Page 5 in the revised manuscript.

Comment 6:   According to which standards were the selected rolling temperatures (25 °C, 300 °C, 400 °C and 500 °C) determined?

Reply 6: Thanks for the reviewer’s suggestion.

We selected the rolling temperatures mainly according to the recrystallization temperature of copper (269 °C). Generally, cold rolling is preformed below the recrystallization temperature while hot rolling is preformed above the recrystallization temperature. Therefore, we choose 25 °C as the temperature parameter for cold rolling and 300 °C, 400 °C and 500 °C for hot rolling to investigate the effect of rolling temperature on the microstructure and property of composite.

The above description is added on Page 7 in the revised manuscript.

Comment 7:  Properties such as the dimensions, purity, and weight ratios of copper and graphene used in the material and method section should be given.

Reply 7: Thanks for the reviewer’s suggestion.

Cu powder was supplied by Ningbo Guangbo Nano Material Co., Ltd, with a purity of 99.9 % and a particle size of about 300 nm. Graphene is synthesized by in situ grown method from oleic acid (OA, over 99.7 % purity, provided by Tianjin Guangfu chemical reagent factory).

The preparation method of graphene/Cu composite is described in ref [10]. Firstly, Cu powder was washed with 0.5 wt.% citric acid aqueous solution to remove the oxide on the Cu particle surface. Secondly, the etched Cu powder slurry was mixed with 0.5 wt.% benzotriazole (BTA) aqueous solution for 1 h and then rinsed with ethanol for 3 times to eliminate the excessive BTA. Thirdly, OA and Cu powder (mass ratio 1:100) were dispersed in the ethanol and kept stirring for 1.5 h. The OA@Cu powder was obtained after drying in vacuum at 80 ℃.

The above description is added on Page 6 in the material and method section of the revised manuscript.

Comment 8:  Giving an experiment schema in the Experimental section is recommended.

Reply 8: Thanks for the reviewer’s suggestion.

We have added an experiment schema (Figure.1) on Page 7 in the Experimental section of the revised manuscript.

Comment 9:  The parameters used in the tensile test should be given in the experimental section.

Reply 9: Thanks for the reviewer’s suggestion.

  In the tensile test, the displacement rate is 0.5 mm/min and the gauge length is 15 mm. The above description is added on Page 8 in the revised Manuscript.

Comment 10:  The resolution of Figure 1 should be increased.

Reply 10: Thanks for the reviewer’s suggestion.

We have increased the resolution of Figure. 1 (Figure. 2 in the revised version) from 150 dpi to 600 dpi. It should be noted that Figs.1a-e (Figs. 2a-e in the revised version) are still not very clear because large step parameter was adopted during EBSD testing.

Comment 11:  The Results section should be discussed further. It should be discussed by comparing it with similar studies in the literature.

Reply 11: Thanks for the reviewer’s suggestion.

According to the reviewer’s comment, we added the MD simulation results to the explanation on the XPS results and the analysis of strengthening mechanisms and fracture morphology. Specific modifications are as follows:

Similar results were also reported previously by molecular dynamics (MD) simulation that a large number of covalent bonds in the graphene/Cu composite were generated during high-temperature hydrostatic compression due to the melting of copper nanoparticles [33]. The above discussion is added on Page 12 in the revised Manuscript.

The major reason is that some early micro-cracks are filled with matrix grains and healed due to the rolling force. Previous molecular dynamics simulation results reported that the copper in plastic flow state is constrained between graphene with self-healing effect [35]. The self-healing effect of copper matrix results in increasing density of the composite [36] and a good interface bonding between graphene and copper matrix. The above discussion is added on Page 17 in the revised Manuscript.

It can be found that the increase of rolling temperature makes the nominal rate of work hardening reduce at the initial rolling stage, which leads to the decrease of the matrix strength. Previous MD simulation results also revealed that graphene boundary could effectively enhance the strength of strain hardening rate of nanocrystalline metals [38]. The above discussion is added on Page 19 and 20 in the revised Manuscript.

Some references for above discussion are listed as follows:

  1. Safina, L. R.; Krylova, K. A.; Baimova, J. A., Molecular dynamics study of the mechanical properties and deformation behavior of graphene/metal composites. Materials Today Physics 2022, 28.
  2. Ma, Y.; Zhang, S.; Xu, Y.; Liu, X.; Luo, S. N., Effects of temperature and grain size on deformation of polycrystalline copper-graphene nanolayered composites. Phys Chem Chem Phys 2020, 22, (8), 4741-4748.
  3. Nakao, W.; Ono, M.; Lee, S.-K.; Takahashi, K.; Ando, K., Critical crack-healing condition for SiC whisker reinforced alumina under stress. Journal of the European Ceramic Society 2005, 25, (16), 3649-3655.
  4. Zhang, S.; Huang, P.; Wang, F., Graphene-boundary strengthening mechanism in Cu/graphene nanocomposites: A molecular dynamics simulation. Materials & Design 2020, 190.

Comment 12:  The article contains numerous typographic and language errors. It should be corrected.

Reply 12: Thanks for the reviewer’s suggestion. We have rechecked the original version and found several typographic and language errors.

(1) In the Introduction Section, we corrected “plastic formation” into “plastic deformation” on page 3 in the revised Manuscript.

(2) In the Materials and Methods Section, we corrected “The rolling process is as follow” into “The rolling process was as follow” on page 6 in the revised Manuscript.

(3) In the Results and Discussion Section, we corrected “the content of C=O bonding show” into “the content of C=O bonding shows” and corrected “increase” into “increases” on page 12 in the revised Manuscript.

Comment 13:  The article should be rearranged by taking into account the journal writing rules and citation rules.

Reply 13: Thanks for the reviewer’s suggestion. We have corrected the format for section title, caption and bibliographic citation. And we believe that the article would be rearranged by editor in detail before publication.

We tried our best to improve the manuscript and made some changes in the manuscript. These changes will not influence the content and framework of the manuscript. We appreciate for Editors/Reviewers’ warm work earnestly, and hope that the correction will meet with approval. Once again, thank you very much for your comments and suggestions.

Yours sincerely,

Yanqiang Liu, Lidong Wang

E-mail: [email protected], [email protected]

Reviewer 2 Report

The manuscript entitled "Effects on the microstructure evolution and properties of graphene/copper composite during rolling process" studies graphene/Cu composites prepared by in situ growth method and rolling treatment. The authors revealed that a high ultimate strength, elongation, and electrical conductivity are attributed to the oriented graphene arrangement and matrix grain refinement. The subject is interesting and the manuscript is well-written and organized. It can be accepted if some comments would be addressed.

Comments:

1) The preparation method of graphene/Cu composite should be described a little bit more, even if it was properly described in [20].

2) Some attention should be paid to the comparison with the simulation results, since from MD simulation it is easier to understand the strengthening mechanisms. For example, refs. [10.1038/s41598-018-21390-1; https://doi.org/10.3390/app13020916;https://doi.org/10.1016/j.matdes.2020.108555; https://doi.org/10.1039/C9CP06830A; https://doi.org/10.1016/j.mtphys.2022.100851]

3) What is the abbreviation GNS in Fig. 2 (Fig. 5, 7)? Is it a single layer of graphene? Or what is the thickness?

Author Response

Dear editors and reviewers: 

Thank you for your letter and the comments concerning our manuscript entitled “Effects on the microstructure evolution and properties of graphene/copper composite during rolling process” (ID: materials-2525325). Those comments are all valuable and very helpful for revising and improving our paper, as well as the important guiding significance to our research. We have studied the comments carefully and have made corrections which we hope to meet with approval.

Revised portions are marked in red in the revised manuscript. The main corrections in the revised manuscript and the responds to the reviewer’s comments are as follows: 

Comment 1:  The preparation method of graphene/Cu composite should be described a little bit more, even if it was properly described in [20].

Reply 1: Thanks for the reviewer’s suggestion.

Cu powder was supplied by Ningbo Guangbo Nano Material Co., Ltd, with a purity of 99.9 % and a particle size of about 300 nm. Graphene is synthesized by in situ grown method from oleic acid (OA, over 99.7 % purity, provided by Tianjin Guangfu chemical reagent factory).

The fabrication process of OA coated Cu powder is as follow: Firstly, Cu powder was washed with 0.5 wt.% citric acid aqueous solution to remove the oxide on the Cu particle surface. Secondly, the etched Cu powder slurry was mixed with 0.5 wt.% benzotriazole (BTA) aqueous solution for 1 h and then rinsed with ethanol for 3 times to eliminate the excessive BTA. Thirdly, OA and Cu powder (mass ratio 1:100) were dispersed in the ethanol and kept stirring for 1.5 h. The OA@Cu powder was obtained after drying in vacuum at 80 ℃。

The above description is added on Page 6 in the revised Manuscript.

Comment 2:  Some attention should be paid to the comparison with the simulation results, since from MD simulation it is easier to understand the strengthening mechanisms. For example, refs. [10.1038/s41598-018-21390-1; https://doi.org/10.3390/app13020916; https://doi.org/10.1016/j.matdes.2020.108555; https://doi.org/10.1039/C9CP06830A; https://doi.org/10.1016/j.mtphys.2022.100851]

Reply 2: Thanks for the reviewer’s suggestion.

According to the reviewer’s comment, we added the MD simulation results to the explanation on the XPS results and the analysis of strengthening mechanisms and fracture morphology. Specific modifications are as follows:

Similar results were also reported previously by molecular dynamics (MD) simulation that a large number of covalent bonds in the graphene/Cu composite were generated during high-temperature hydrostatic compression due to the melting of copper nanoparticles [33]. The above discussion is added on Page 12 in the revised Manuscript.

The major reason is that some early micro-cracks are filled with matrix grains and healed due to the rolling force. Previous molecular dynamics simulation results reported that the copper in plastic flow state is constrained between graphene with self-healing effect [35]. The self-healing effect of copper matrix results in increasing density of the composite [36] and a good interface bonding between graphene and copper matrix. The above discussion is added on Page 17 in the revised Manuscript.

It can be found that the increase of rolling temperature makes the nominal rate of work hardening reduce at the initial rolling stage, which leads to the decrease of the matrix strength. Previous MD simulation results also revealed that graphene boundary could effectively enhance the strength of strain hardening rate of nanocrystalline metals [38]. The above discussion is added on Page 19 and 20 in the revised Manuscript.

Some references for above discussion are listed as follows:

  1. Safina, L. R.; Krylova, K. A.; Baimova, J. A., Molecular dynamics study of the mechanical properties and deformation behavior of graphene/metal composites. Materials Today Physics 2022, 28.
  2. Ma, Y.; Zhang, S.; Xu, Y.; Liu, X.; Luo, S. N., Effects of temperature and grain size on deformation of polycrystalline copper-graphene nanolayered composites. Phys Chem Chem Phys 2020, 22, (8), 4741-4748.
  3. Nakao, W.; Ono, M.; Lee, S.-K.; Takahashi, K.; Ando, K., Critical crack-healing condition for SiC whisker reinforced alumina under stress. Journal of the European Ceramic Society 2005, 25, (16), 3649-3655.
  4. Zhang, S.; Huang, P.; Wang, F., Graphene-boundary strengthening mechanism in Cu/graphene nanocomposites: A molecular dynamics simulation. Materials & Design 2020, 190.

Comment 3:  What is the abbreviation GNS in Fig. 2 (Fig. 5, 7)? Is it a single layer of graphene? Or what is the thickness?

Reply 3: Thanks for the reviewer’s suggestion.

We have rechecked the original version and found the mistake that we did not give the full name of GNS. Actually, GNS is the abbreviation of graphene nanosheet, which is considered as a general term of few-layer (no more than 10 layers) graphene and multi-layer (10-30 layers). The thickness of graphene nanosheet is between 0.335 nm to 100 nm. The full name of GNS is given on Page 9 in the revised Manuscript.

We tried our best to improve the manuscript and made some changes in the manuscript. These changes will not influence the content and framework of the manuscript. We appreciate for Editors/Reviewers’ warm work earnestly, and hope that the correction will meet with approval. Once again, thank you very much for your comments and suggestions.

Yours sincerely,

Yanqiang Liu, Lidong Wang

E-mail: [email protected], [email protected]

Reviewer 3 Report

The research article by Z. Yang focuses on the study of graphene/Cu composites during the rolling deformation at different rolling temperature and provides detailed investigations about mechanical and electrical properties of the composites. The main morphological effects, such as microstructures evolution, have been systematically investigated, with a particular attention to the cold-rolled composite about which poor is known. In general, the manuscript is well-written and turns out to be potentially useful for experts in the field and / or for all the scientist approaching the use of graphene as a filler to built composite materials which have to face with rolling phenomena and adhesion between graphene and Cu. Nevertheless, in some points the discussion could be broadened to increase the readability and references are missing or can be increased; especially the introduction, could be improved by referring to the uniqueness of  graphene-based composite materials compared to composites containing other 2D materials. Some other aspects need to be clarified before considering the article suitable for publication. Recommendations to improve the quality of the paper are listed below:

- lines 29-30: authors introduce graphene/metal composites mentioning huge potential in thermal management, electrical contact and cable applications. Some aspects faced in the article, including the distribution of graphene sheets along Cu grain boundaries/the diffusion to grain inner or the adhesion between graphene and Cu are crucial issues not only for design of composites materials but also for graphene synthesis by CVD which generally is preformed using thin-metal films and in particular Cu substrate. I believe that focusing on this aspect can improve the introduction, attracting readers even outside the production of graphene-based composite materials. The authors can critically use the following references (Carbon 2014 - https://doi.org/10.1016/j.carbon.2013.11.020; Cell Report Physical Science 2021 - https://doi.org/10.1016/j.xcrp.2021.100372;)

- In the introduction, authors often mention the mechanical properties of graphene/metal composites; With the aim to emphasize the results obtained with composite materials, I think the authors should mention relatively recent works that have deeply investigated the mechanical properties of deformed graphene (authors can critically use the following works Small 2021 https://doi.org/10.1002/smll.202104487; Phys. Rev. Materials 2023 https://doi.org/10.1103/PhysRevMaterials.7.054007;). Even a brief mention of such aspects can strengthen the benefits of using graphene-based composites with respect to other 2D materials-based composites.

- at lines 106 authors state: “…the cold-rolled pure copper consists of ~2 µm equiaxed grains”. Are the dimensions coherent with values found in literature? Figure 2a shows some grains even smaller than 1µm; please clarify.

- at line 108 authors state: “…the grains have … a thickness of only 200 nm” Where does the thickness value come from? in the present form it seems from the figure, if it does not come from the figure the sentence have to be modified for clarity.

- at lines 110-111 authors state: “…graphene sheets are distributed in the Cu grain boundary and even directionally arranged along the rolling direction without any agglomeration”; again here it seems the results about the graphene sheets distribution arise directly from the figure 2 but , if I well understood, graphene is not deductible from the EBSD; please clarify.

- at lines 137-138 authors state:The reason may be that high rolling temperature can accelerate copper surface catalysis and reduce graphene defects.” At least a reference should be added to strengthen the speculation.

- The caption of Figure5 should be implemented; caption of arrows (white and yellow) as well as for yellow dashed circles, have to be added.

- at lines 189-190 authors state:”…except the composite with 30% deformation…” for completeness, few comments about the worsening of strength and elongation % occurring at composite with 30% deformation should be added.

- at line 230 authors attribute to three factor the impacts of rolling temperature on the mechanical properties. In the present form, the first and the second factor are both attributable to the role of the rolling temperature which in turns affect the Cu grain size and the graphene distribution. I think this point could be rephrased, for clarity

Minor

-line 92. Electron backscattering diffraction should be added not only in the acronym form EBSD. This could help the reading as well of people out of the topic.

Author Response

Dear editors and reviewers: 

Thank you for your letter and the comments concerning our manuscript entitled “Effects on the microstructure evolution and properties of graphene/copper composite during rolling process” (ID: materials-2525325). Those comments are all valuable and very helpful for revising and improving our paper, as well as the important guiding significance to our research. We have studied the comments carefully and have made corrections which we hope to meet with approval.

Revised portions are marked in red in the revised manuscript. The main corrections in the revised manuscript and the responds to the reviewer’s comments are as follows: 

Comment 1: lines 29-30: authors introduce graphene/metal composites mentioning huge potential in thermal management, electrical contact and cable applications. Some aspects faced in the article, including the distribution of graphene sheets along Cu grain boundaries/the diffusion to grain inner or the adhesion between graphene and Cu are crucial issues not only for design of composites materials but also for graphene synthesis by CVD which generally is preformed using thin-metal films and in particular Cu substrate. I believe that focusing on this aspect can improve the introduction, attracting readers even outside the production of graphene-based composite materials. The authors can critically use the following references (Carbon 2014 - https://doi.org/10.1016/j.carbon.2013.11.020; Cell Report Physical Science 2021 - https://doi.org/10.1016/j.xcrp.2021.100372;)

Reply 1: Thanks for the reviewer’s suggestion.

According to reviewer’s comment, we have added research background about high-quality graphene synthesis into introduction section and cited the two suggested references. Specific modifications are as follows.

……Moreover, high-quality graphene grown on Cu substrate or thin-metal films by chemical vapor deposition (CVD) method is also expected to be used in electronic devices [10, 11]. The common problems that researchers need to face come into three aspects: (1) graphene easily aggregates in the metal matrix due to its huge surface energy and poor wettability to metals, which leads to overall performance degradation [12, 13]; (2) the fabrication process may destroy graphene microstructure, which directly causes the fall in electrical or thermal properties of graphene or its composites [14-16]; (3) the adhesion between graphene and Cu is usually poor.

……This work would provide a better understanding of the rolling behavior of graphene/Cu composite and offer a novel strategy for the production of high-quality graphene and graphene-based composite.

The above description on Page 3 and 5 in the revised Manuscript.

Comment 2: In the introduction, authors often mention the mechanical properties of graphene/metal composites; With the aim to emphasize the results obtained with composite materials, I think the authors should mention relatively recent works that have deeply investigated the mechanical properties of deformed graphene (authors can critically use the following works Small 2021 https://doi.org/10.1002/smll.202104487; Phys. Rev. Materials 2023 https://doi.org/10.1103/PhysRevMaterials.7.054007;). Even a brief mention of such aspects can strengthen the benefits of using graphene-based composites with respect to other 2D materials-based composites.

Reply 2: Thanks for the reviewer’s suggestion.

According to reviewer’s comment, we have added the investigation about the mechanical properties of deformed graphene into introduction section. Specific modifications are as follows.

…Besides graphene/metal composites, graphene also exhibits its own distinctive property in the case of deformation. Mescola et al deposited graphene over different nanotextured silicon substrates and found that asymmetric straining of C-C bonds over the corrugated architecture resulting in distinct friction dissipation [25,26]. Thus, graphene deposited onto an anisotropic nanotextured system could acquire diverse nanomechanical properties, which haven’t been found in other 2D materials.

The above description on Page 4 in the revised Manuscript. Some references for above discussion are listed as follows:

[25] Mescola, A.; Paolicelli, G.; Ogilvie, S. P.; Guarino, R.; McHugh, J. G.; Rota, A.; Iacob, E.; Gnecco, E.; Valeri, S.; Pugno, N. M.; Gadhamshetty, V.; Rahman, M. M.; Ajayan, P.; Dalton, A. B.; Tripathi, M., Graphene Confers Ultralow Friction on Nanogear Cogs. Small 2021, 17, (47), e2104487.

[26] Mescola, A.; Silva, A.; Khosravi, A.; Vanossi, A.; Tosatti, E.; Valeri, S.; Paolicelli, G., Anisotropic rheology and friction of suspended graphene. Physical Review Materials 2023, 7, (5).

Comment 3: at lines 106 authors state: “…the cold-rolled pure copper consists of ~2 µm equiaxed grains”. Are the dimensions coherent with values found in literature? Figure 2a shows some grains even smaller than 1µm; please clarify.

Reply 3: Thanks for the reviewer’s suggestion.

We have corrected the description from “…the cold-rolled pure copper consists of ~2 µm equiaxed grains” to “…the cold-rolled pure copper consists of 0.8-2 µm equiaxed grains” on Page 9 in the revised Manuscript.

Comment 4: at line 108 authors state: “…the grains have … a thickness of only 200 nm” Where does the thickness value come from? in the present form it seems from the figure, if it does not come from the figure the sentence has to be modified for clarity.

Reply 4: Thanks for the reviewer’s suggestion.

We have corrected the description from “…the grains have … a thickness of only 200 nm” to “…the grains have … a thickness of only 150-300 nm as shown in Fig 2b” on Page 9 in the revised Manuscript.

Comment 5: at lines 110-111 author states: “…graphene sheets are distributed in the Cu grain boundary and even directionally arranged along the rolling direction without any agglomeration”; again here it seems the results about the graphene sheets distribution arise directly from the figure 2 but, if I well understood, graphene is not deductible from the EBSD; please clarify.

Reply 5: Thanks for the reviewer’s suggestion.

We have rechecked the original version and deleted the misleading sentence “Moreover, graphene is distributed in the Cu grain boundary and even directionally arranged along the rolling direction without any agglomeration” on Page 9 in the revised Manuscript.

Comment 6: at lines 137-138 authors state: “The reason may be that high rolling temperature can accelerate copper surface catalysis and reduce graphene defects.” At least a reference should be added to strengthen the speculation.

Reply 6: Thanks for the reviewer’s suggestion.

We have added a reference [31] [Zhang, X.; Shi, C. S.; Liu, E. Z.; Zhao, N. Q.; He, C. N., Effect of Interface Structure on the Mechanical Properties of Graphene Nanosheets Reinforced Copper Matrix Composites. Acs Applied Materials & Interfaces 2018, 10, (43), 37586-37601]to prove the effect of temperature on the catalytic ability of copper and strengthen our speculation on Page 11 in the revised Manuscript.

Comment 7: The caption of Figure5 should be implemented; caption of arrows (white and yellow) as well as for yellow dashed circles, have to be added.

Reply 7: Thanks for the reviewer’s suggestion.

We have added “The locations of the yellow arrows in (a) and yellow dashed circles in (b-d) show the graphene and the white arrows mark Cu twins in the composite.” in the caption of Figure 6 on Page 15 in the revised Manuscript.

Comment 8: at lines 189-190 authors state:”…except the composite with 30% deformation…” for completeness, few comments about the worsening of strength and elongation % occurring at composite with 30% deformation should be added.

Reply 8: Thanks for the reviewer’s suggestion.

The main reason that the strength of the composite with 30% rolling reduction decreases may be that the rotation of graphene cannot coordinate with the deformation of Cu grains which results in crack initiation at the early stage of cold rolling process. The above comment on Page 15 in the revised Manuscript.

Comment 9: at line 230 authors attribute to three factor the impacts of rolling temperature on the mechanical properties. In the present form, the first and the second factor are both attributable to the role of the rolling temperature which in turns affect the Cu grain size and the graphene distribution. I think this point could be rephrased, for clarity

Reply 9: Thanks for the reviewer’s suggestion.

We have rechecked the original version and found that our previous expression is inaccuracy. According to reviewer’s comment, we rephrased the corresponding discussion section as follows.

The impacts of rolling temperature on the mechanical properties of the graphene/Cu composites can be attributed to grain refinement strengthening, grain boundary strengthening and working hardening. Firstly, the grain refinement strengthening mechanism can be expressed by:

                                                          σ =σ0+k*d-1/2 (1)

where σ is the yield strength of the composite,  σ0 is a frictional stress required to move dislocations, k is the Hall-Petch slope and d is the grain size. According to the above TEM result, the rolling temperature has a remarkable influence on the grain size of Cu matrix and an obvious grain refinement can be achieved by cold rolling treatment. Secondly, the rolling temperature also affects grain boundary strengthening in the composite…

The above discussion on Page 18 and 19 in the revised Manuscript.

We tried our best to improve the manuscript and made some changes in the manuscript. These changes will not influence the content and framework of the manuscript. We appreciate for Editors/Reviewers’ warm work earnestly, and hope that the correction will meet with approval. Once again, thank you very much for your comments and suggestions.

Yours sincerely,

Yanqiang Liu, Lidong Wang

E-mail: [email protected], [email protected]

Round 2

Reviewer 1 Report

The authors have completed the necessary revisions. This article can be accepted for publication in its final form.

Reviewer 3 Report

The authors addressed all the points raised-up in the first round revision.